# Assessment and site-specific manipulation of DNA (hydroxy-)methylation during mouse corticogenesis

Florian Noack[1] , Abhijeet Pataskar[2], Martin Schneider[3], Frank Buchholz[3], Vijay K Tiwari[2], Federico Calegari[1]

**Dynamic changes in DNA (hydroxy-)methylation are fundamental for stem cell differentiation. However, the signature of these epigenetic marks in specific cell types during corticogenesis is unknown. Moreover, site-specific manipulation of cytosine modifications is needed to reveal the significance and function of these changes. Here, we report the first assessment of (hydroxy-)methylation in neural stem cells, neurogenic progenitors, and newborn neurons during mammalian corticogenesis. We found that gain in hydroxymethylation and loss in methylation occur sequentially at specific cellular transitions during neurogenic commitment. We also found that these changes predominantly occur within enhancers of neurogenic genes up-regulated during neurogenesis and target of pioneer transcription factors. We further optimized the use of dCas9-Tet1 manipulation of (hydroxy-)methylation, locus-specifically, in vivo, showing the biological relevance of our observations for *Dchs1*, a regulator of corticogenesis involved in developmental malformations and cognitive impairment. Together, our data reveal the dynamics of cytosine modifications in lineage-related cell types, whereby methylation is reduced and hydroxymethylation gained during the neurogenic lineage concurrently with up-regulation of pioneer transcription factors and activation of enhancers for neurogenic genes.**

## Introduction

During embryonic development of the mammalian brain, neural stem and progenitor cells progressively switch from proliferative to differentiative divisions to generate neurons and glia that populate the cortical layers. This switch ultimately controls cortical thickness and surface area during development and evolution and is important for better understanding the molecular cell biology of somatic stem cells and their potential use in therapy (LaMonica et al, 2012; Southwell et al, 2014; Taverna et al, 2014). Although it is increasingly clear that cell fate commitment involves a massive

rewiring of gene expression programs, the epigenetic mechanisms underlying these changes are yet to be fully understood.

Specifically, before neurogenesis, radial glial, neural stem cells populating the ventricular zone (VZ) undergo symmetric, proliferative divisions that exponentially expand their number over time. With development, an increasing proportion of radial glial cells switches to asymmetric, differentiative divisions to generate basal, intermediate progenitors that leave the VZ to form the subventricular zone (SVZ) (Lui et al, 2011; Taverna et al, 2014). Although the majority (~80%) of basal progenitors are soon consumed to generate neurons, a subpopulation remains that undergoes a few rounds of symmetric proliferative divisions to expand their pool within the SVZ (Lui et al, 2011; Taverna et al, 2014). As a result, both proliferative progenitors (PPs) and differentiative progenitors (DPs) coexist as intermingled populations in the two germinal zones of the mammalian VZ and SVZ, whereas neurons are added to the cortical layers.

Understanding the switch from proliferation to differentiation of neural progenitors, thus, requires the identification of PP from DP and neurons. To this end, our group has previously characterized a double reporter mouse line that allowed the isolation of DP and neurons by the expression of $Btg2^{RFP}$ and $Tubb3^{GFP}$, respectively, and of PP by the lack of both reporters (Aprea et al, 2013). This approach led to the identification of genes transitorily up- or down-regulated specifically in DP relative to both PP and neurons and, thus, identifying the functional signature to neurogenic commitment (Aprea et al, 2015; Aprea et al, 2013; Artegiani et al, 2015; de Jesus Domingues et al, 2016). To further decipher the regulatory mechanisms underlying changes in gene expression, we here decided to characterize the epigenetic signature of PP, DP, and neurons and its role in cortical development.

In recent years, epigenetic modifications have emerged as a key mechanism regulating gene expression during differentiation of embryonic and adult neural stem cells (Hirabayashi & Gotoh, 2010; Smith & Meissner, 2013; Yao et al, 2016). Among these, modification of DNA, in particular cytosine methylation and hydroxymethylation, was shown to be critical for cell fate change in the developing and adult mammalian brain (Koh & Rao, 2013; Smith & Meissner, 2013; Schubeler, 2015; Stricker & Götz, 2018). For example, deletion of

---

[1]CRTD-Center for Regenerative Therapies, School of Medicine, Technische Universität Dresden, Dresden, Germany    [2]Institute of Molecular Biology, Mainz, Germany    [3]Medical Systems Biology, School of Medicine, Technische Universität Dresden and Max Planck Institute for Molecular Cell Biology and Genetics, Dresden, Germany

Correspondence: federico.calegari@tu-dresden.de

methyltransferases (*Dnmt1 or 3a*) impaired neuronal differentiation resulting in a reduced number of neurons (Fan et al, 2005; Nguyen et al, 2007; Wu et al, 2010). Similarly, manipulating the ten-eleven translocation genes (*Tet1 or 3*), which encode enzymes oxidizing 5-methylcytosine (5mC) to 5-hydroxymethylcytosine (5hmC) driving demethylation (Tahiliani et al, 2009; He et al, 2011), led to altered corticogenesis (Hahn et al, 2013), neuronal death (Xin et al, 2015), aberrant circuitry formation (Zhu et al, 2016), impaired adult neurogenesis, and reduced cognitive performance (Kaas et al, 2013; Rudenko et al, 2013; Zhang et al, 2013). However, as these attempts to manipulate DNA methylation were limited to systemic genetic deletions or pharmacological treatments causing genome-wide changes, they failed to provide functional information with regard to locus-specific effects. Moreover, current efforts to understand the dynamic changes in (hydroxy-)methylcytosine during brain development have so far been limited to in vitro models of neurogenesis or bulk tissues (Stadler et al, 2011; Hahn et al, 2013; Lister et al, 2013; Mo et al, 2015; Ziller et al, 2015; Jaffe et al, 2016).

To fill these gaps and decipher the role of DNA (de-)methylation during brain development, systems are required that allow the (i) identification of 5mC and 5hmC (together referred to as 5(h)mC) patterns in specific cell types, rather than whole tissues, during neurogenesis and (ii) efficient manipulation of DNA (hydroxy-)methylation in a locus-specific, rather than systemic, manner for which approaches with varying efficiency were recently described (Liu et al, 2016; Morita et al, 2016). In this work, we achieved both providing the first cell type–specific resource of 5(h)mC patterns in PP, DP, and neurons. This revealed that a commitment to neurogenesis in DP is characterized by an increase in 5hmC within enhancers of neurogenic genes and resulting in the subsequent loss in 5mC in their neuronal progeny. We also found that these changes correlated with the (i) acquisition of histone marks characteristic of open chromatin, (ii) up-regulation and putative binding of basic helix-loop-helix (bHLH) pioneer transcription factors, and (iii) activation of their nearby neurogenic genes. Finally, we optimized the use of dCas9-Tet1 to achieve the locus-specific manipulation of (hydroxy-)methylation in developing mouse embryos and provided one example of the potential biological relevance of our observations for one candidate 5(h)mC-dependent regulator of neurogenesis involved in neurodevelopmental malformations and mental retardation: *Dchs1*.

## Results

### Cell-specific assessment of 5(h)mC identifies the bivalent epigenetic signature of enhancers during neurogenic commitment

To characterize the signature of DNA (hydroxy-)methylation during corticogenesis, we FAC-sorted PP, DP, and neurons from *Btg2*$^{RFP}$/*Tubb3*$^{GFP}$ mice by their combinatorial expression of the two fluorescent reporters (RFP–/GFP–, RFP+/GFP–, and GFP+ irrespective of RFP, respectively) (Aprea et al, 2013). Mice at the embryonic (E) day 14.5 were chosen as a stage at mid-corticogenesis, that is, when the three cell types were similarly abundant. We next generated genome-wide maps of both 5mC and 5hmC by adapting a comparative immunoprecipitation protocol (Tan et al, 2013) by which DNA from different cell types were barcoded, mixed, and immunoprecipitated together considerably reducing experimental variability among samples. Antibodies against 5mC or 5hmC were used and single-end sequencing performed in three biological replicates (~35–60 million unique reads per sample) to cell-specifically assess the (hydroxy-)methylome of each cell type (Figs 1A and S1A).

After assessing the quality and specificity of our immunoprecipitation (Fig S1B), we started to validate our approach by merging together PP, DP, and neurons and comparing our data with previous studies performed in cell cultures or bulk brain tissues (Hahn et al, 2013; Oda et al, 2006; Song et al, 2011; Tan et al, 2013; Wen & Tang, 2014). This confirmed the general hyper- or hypo-(hydroxy-)methylation state of previously reported individual loci, including *Xist*, *Hist1h2aa*, *Gapdh*, *Uchl1*, and others, which in all cases showed relative levels of 5(h)mC consistent with previous studies (Fig S1C and D). Next, we assessed the genome-wide distribution of 5(h)mC across gene bodies. In doing so, we also sought to investigate a potential correlation among (hydroxy-)methylation patterns of differentially expressed genes. Toward this, we selected from the expression profile of the same cell types previously reported by our group (Aprea et al, 2013, 2015) the top 10% most highly or lowly expressed genes (FPKM ≥ 2) and plotted the levels of 5(h)mC across the two groups. This confirmed the characteristic (Song et al, 2011; Hahn et al, 2013) drop in 5(h)mC at the transcription start and end sites (TSS and TES) (Fig 1B). In addition, we found that highly expressed genes had lower 5mC levels specifically across their TSS and the first third of the gene body, whereas the inverse correlation was found toward the 3′ end (Fig 1B, top). A similar trend, but limited to a smaller region within the TSS, was detected for 5hmC, whereas, conversely, higher levels correlated with increased expression across other regions of the gene body (Fig 1B, bottom) (similar trends were also observed when analyzing individual cell types, not shown).

Additional analyses of 5(h)mC patterns across several genomic features, including exons, introns, CpG islands, and enhancers (according to Ensembl annotation) of the E14.5 mouse revealed that the most significant enrichment in cytosine modifications, particularly 5hmC, occurred within enhancers (Fig 1C, note that in these coverage-profiles, the variance was too small to be depicted). Interestingly, whereas these analyses pointed out the distribution of 5(h)mC within genomic features when pulling cell types together, analysis of PP, DP, and neurons individually revealed a general decrease in 5mC and increase in 5hmC during differentiation that were particularly profound within enhancers relative to other regions (Fig 1D and data not shown). Specifically, a loss in 5mC within enhancers was primarily observed in the transition from DP to neurons but not from PP to DP (Fig 1D, top), whereas a constant increase in 5hmC was observed during neurogenic commitment from PP to DP as well as from DP to neurons (Fig 1D, bottom). Knowing that a gain in 5hmC by Tet enzymes is essential for active demethylation at enhancers (Hon et al, 2014; Lu et al, 2014), this suggested that the bivalent epigenetic signature within enhancers of DP, with a gain in 5hmC but not yet a loss in 5mC, potentially represented an initial step in the transition from high 5mC in PP and its subsequent loss in neurons. This in turn implied that neurogenic

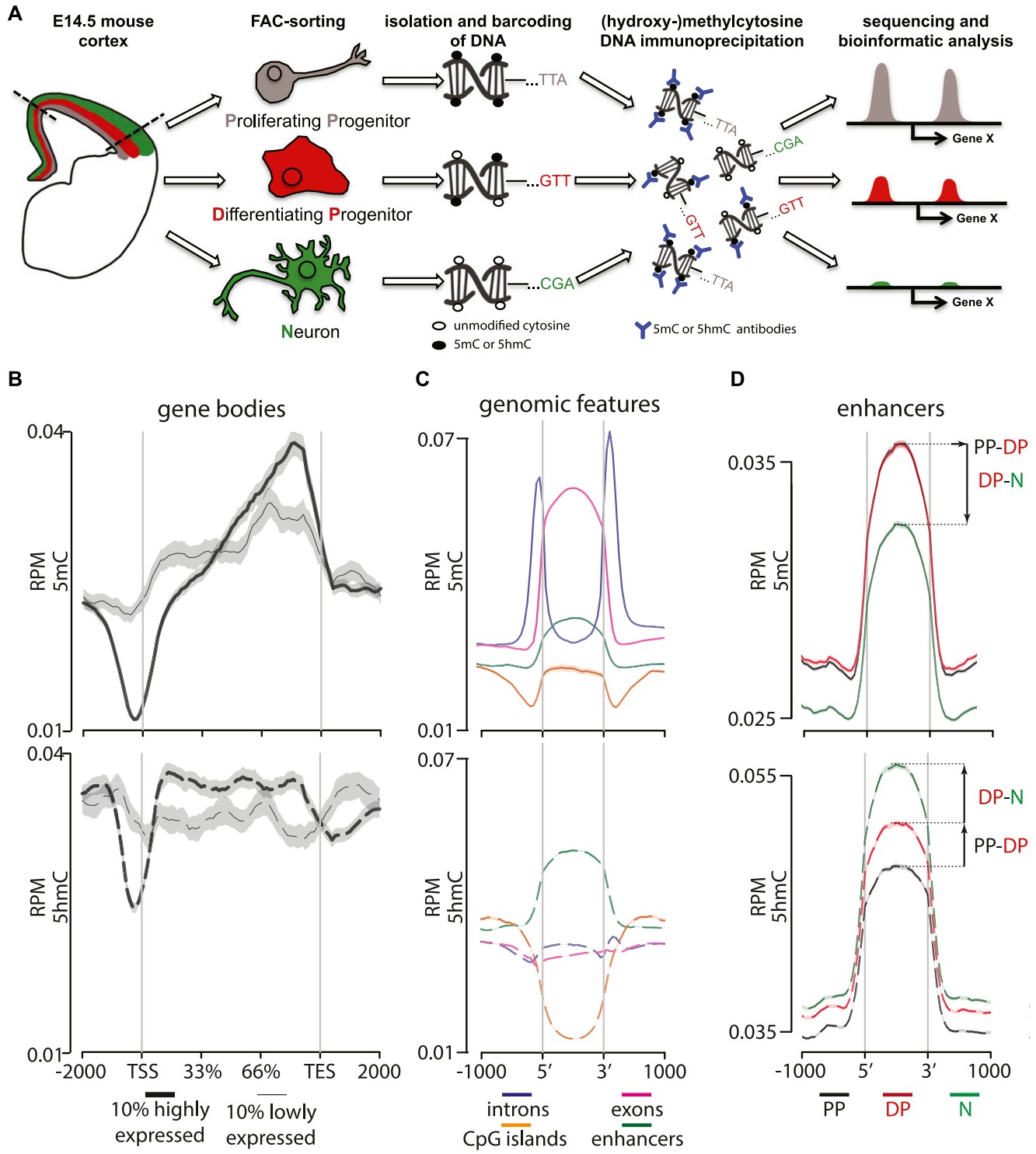

**Figure 1. Cell type–specific assessment of 5(h)mC.**
**(A)** Drawing depicting the strategy of our study with PP, DP, and neurons (color coded in gray, red, and green, respectively, throughout all figures) FAC-sorted from the E14.5 mouse cortex followed by DNA barcoding, mixture, immunoprecipitation, and sequencing. **(B–D)** Distribution and abundance (reads per million = RPM) of 5(h)mC of the ±2,000-bp regions across gene bodies of the top 10% more highly or lowly expressed genes (B; thick and thin lines, respectively); genomic regions (C, as indicated); or enhancers in PP, DP, and neurons (D, as indicated). Note that in this and all other figures, 5mC and 5hmC are consistently depicted as continuous (top) or dashed (bottom) lines, respectively. Values for the three cell types were merged together (B, C) or depicted individually (D). Note the characteristic drop in 5(h)mC at the TSS and TES (B) and the bivalent 5(h)mC signatures within enhancers during neurogenic commitment (D). Gray shadows in B represent the standard error of the mean, which was too small to be visibly depicted in (C, D) because of the much higher number of loci in the latter relative to the former plots.

commitment involves stage-specific, active remodeling of DNA methylation at distal regulatory elements that may have escaped previous analyses of bulk tissues or cell lines. In addition, our data identified DP as a bivalent population defined by levels of 5mC similar to their precursors PP but intermediate in 5hmC between PP and their neuronal progeny.

### A concurrent loss in 5mC and gain in 5hmC occurs within, or in proximity to, neurogenic genes up-regulated during fate commitment

We then identified the loci whose 5(h)mC levels changed (±50%; $P <$ 0.05) in specific cell types. Altogether, this led to ~25,000 differentially (hydroxy-)methylated loci associated to 11,000 unique genes (nearest TSS) in the PP-DP or DP-neuron transition (Fig 2A and Supplemental Data 1 and 2).

Interestingly, our analysis showed that the number of loci losing 5mC (5mC-loss) during the PP-DP (1,927) or DP-neuron (3,716) transition was overall 1.3- and 2.5-fold greater than the number of loci undergoing the converse modification and gain in 5mC (5mC-gain: 1,475 and 1,463 for PP-DP and DP-neurons, respectively) (Fig 2A, left: compare loss versus gain, orange and blue, respectively). Concomitantly, a ~3.0- and 2.0-fold greater number of loci gained 5hmC (5hmC-gain) relative to those losing it (5hmC-loss) during each cellular transition (4,757 versus 1,611 and 6.051 versus 3.043, respectively) (Fig 2A, right: compare loss versus gain, orange and blue, respectively). The trend and magnitude of these changes were reminiscent of the bivalent (hydroxy-)methylation signature of DP within enhancers (Fig 1D) with the greatest loss in 5mC occurring primarily during the DP–neuron transition, whereas an increase in 5hmC was similarly abundant during both transitions (Fig 2A). In addition, the overall >2-fold greater abundance in 5mC-loss and 5hmC-gain loci relative to the converse modifications (i.e., 5mC-gain and 5hmC-loss) suggested a functional role for demethylation in neurogenic commitment. However, these trends were insufficient to ascertain whether a loss in 5mC and gain in 5hmC occurred within the same, rather than different, loci and, if so, whether changes occurred at the level of the same, rather than different, cytosines.

To address the first question, we selected 5hmC-gain loci at the onset of neurogenic commitment in DP (4,757) and assessed within these loci the changes in 5mC (Fig 2B). Conversely, we also tested the inverse correlation after selecting all 5mC-loss loci in neurons (3,716) and measuring within them the levels of 5hmC as before (Fig 2C). In both cases, we found a highly significant ($P < 1 \times 10^{-20}$) negative correlation both at the level of individual loci and mean 5(h)mC values (Fig 2B and C, heat and violin plots, respectively), indicating that in the overwhelming majority of the cases, both modifications occurred concomitantly within the same, 5mC-loss/5hmC-gain, loci. Showing the specificity of these results, a much weaker, if any, correlation was found among 5hmC-loss (1,611) and 5mC-gain (1,463) loci (Fig S2A and B), indicating that 5mC-loss/5hmC-gain and 5mC-gain/5hmC-loss loci are functionally distinct.

We next selected six among this pool of 5mC-loss/5hmC-gain loci, each containing 5–10 individual cytosines (48 in total) and performed both bisulfite and oxidative bisulfite amplicon sequencing from genomic DNA of PP, DP, and neurons. This was important not only to validate our (h)MeDIP analysis at single-nucleotide resolution but also as a means to investigate whether changes in 5(h)mC occurred at the level of the same cytosines rather than different nucleotides within the same locus. For all six loci, this not only validated the relative levels of 5(h)mC previously assessed by (h)MeDIP but also showed that in essentially all cases (44/48 cytosines; i.e., >90%), a 5mC-loss/5hmC-gain involved the same cytosine residues in subsequent cellular transitions (three examples are shown in Fig S2C). In addition, and confirming previous results, the magnitude of the loss in 5mC from PP to DP, if any, was typically minor (on average ~10%) and only became substantial from DP to neurons (50% decrease), whereas the magnitude of a gain in 5hmC was more robust and similar (twofold increase) in both cellular transitions (Figs 2D and S2C).

To gain insight into the biological role of loci undergoing differential (hydroxy-)methylation, we next investigated genes associated with 5mC-loss/5hmC-gain loci (nearest TSS). Gene ontology (GO) term analysis revealed a highly consistent and very strong enrichment in neurogenesis-related terms, including as the top three most enriched: *neurogenesis*, *developmental process*, and *cell signaling* (Fig 2E). Highlighting the specificity of 5mC-loss/ 5hmC-gain loci in neurogenic commitment, genes containing, or in proximity to, loci undergoing the converse modifications of 5mC-gain/5hmC-loss showed less consistent terms with lower enrichment scores and that in some cases were not even specific for neuronal development, such as *mesonephros development* and *metabolism* (Fig S2D). Finally, again taking advantage of the previous assessment of gene expression in PP, DP, and neurons of the E14.5 mouse cortex (Aprea et al, 2013, 2015), we reconstructed the expression levels of transcripts associated with 5mC-loss/5hmC-gain loci finding a highly significant up-regulation in the lineage from PP to DP and reaching the higher expression levels in neurons (Fig 2F). In turn, this indicated that 5hmC-gain alone correlated with increased gene expression even before a loss in 5mC. Conversely, no correlation was detected among genes associated with 5mC-gain/5hmC-loss loci (Fig S2E).

Altogether, our study revealed the sequential molecular changes in 5(h)mC in consecutive cell divisions during the neurogenic lineage whereby hydroxymethylation is initiated at the onset of fate commitment in DP and resulting in the subsequent loss in methylation in their neuronal progeny. Such 5mC-loss/5hmC-gain was more prevalent in loci annotated as enhancers and located within, or in proximity to, neurogenic genes up-regulated during differentiation, suggesting that active demethylation is a trigger of neurogenesis and establishment of neuronal identity.

### 5mC-loss/5hmC-gain loci are enriched in active enhancers and neurogenic pioneer factor–binding motifs

We next inspected the features of 5mC-loss/5hmC-gain loci. Extending our previous analysis at the genome-wide level (Fig 1D), we also confirmed among 5mC-loss/5hmC-gain loci a high enrichment in Ensembl-annotated enhancers (average z-score 41.56; $P < 0.001$). To investigate whether these enhancers were characterized by an active chromatin state, we took advantage of the large depository of ChIP-seq datasets provided by the Encyclopedia of DNA Elements (ENCODE) (Shen et al, 2012). To this end, we overlaid

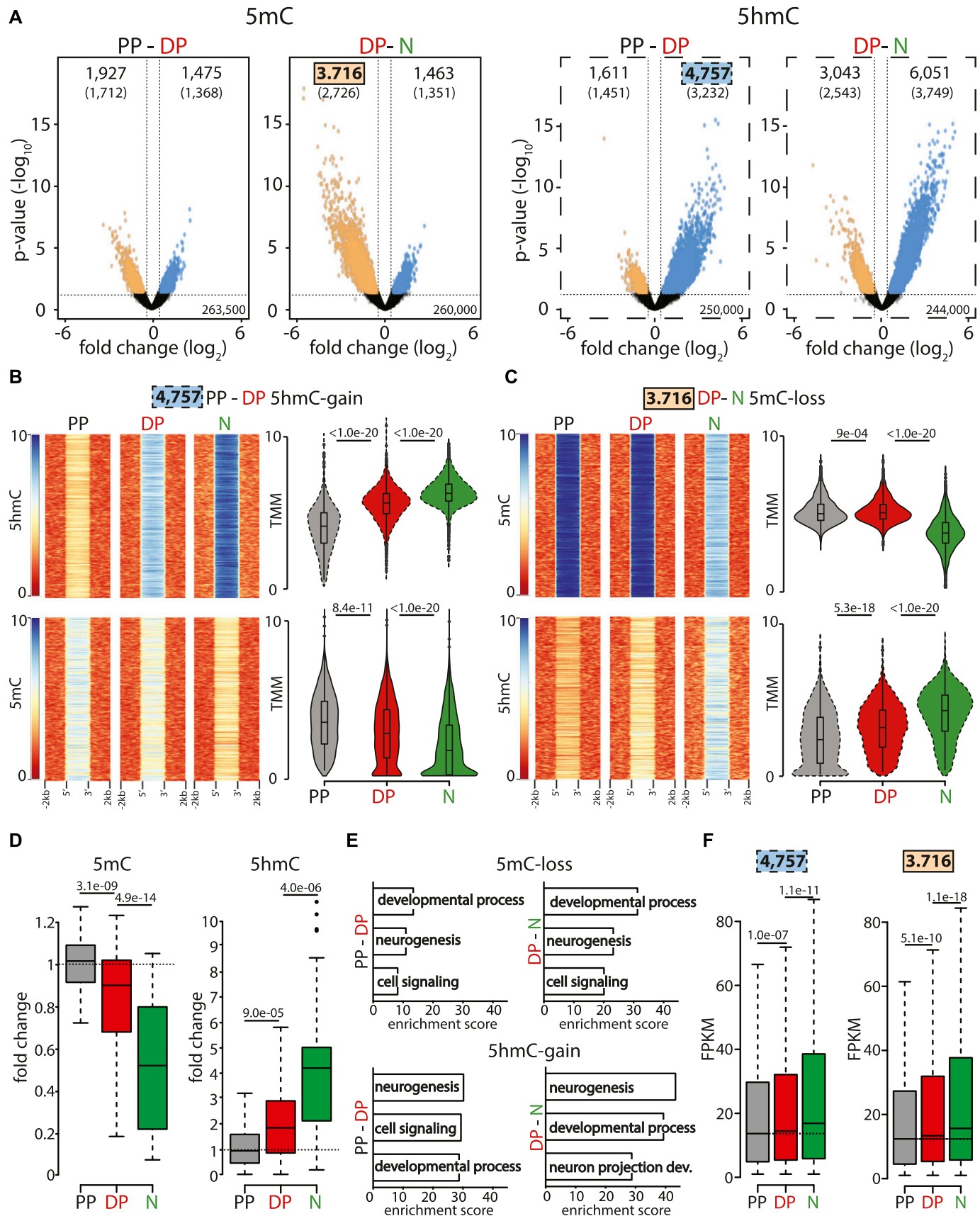

our differentially (hydroxy-)methylated loci with that of histone H3 lysine 27 acetylation (H3K27ac) and/or lysine 4 methylation (H3K4me1) ChIP-seq data as signatures of active and poised enhancers, respectively (Mahe et al, 2017). By specifically selecting data obtained from the E14.5 mouse cerebral cortex as equivalent to our study, we found that 5mC-loss/5hmC-gain loci displayed a sharp increase at their 5′ and 3′ end sites in H3K27ac as well as H3K4me1 levels relative to both 5mC-gain/5hmC-loss and random genomic loci (Figs 3A and S3A).

Given the strong correlation of 5mC-loss/5hmC-gain loci with increased gene expression (Fig 2F) and enrichment in active enhancers (Fig 3A), we inspected whether these loci additionally contained distinctive sequence motifs. We found a remarkably high enrichment in binding motifs for hundreds of transcription factors with the most significant one belonging to the bHLH family and that were very distinct from the binding motifs found within 5mC-loss/5hmC-gain loci (Fig S3B, top five motifs). However, we noticed that such a long list of bHLH transcription factors also included genes that according to our previous transcriptome analysis of PP, DP, and neurons (Aprea et al, 2013, 2015) were either expressed at negligible levels at this developmental stage or not expressed at all (e.g., Bhlha15, Twist1, Atoh1, and others). This in turn suggested that our motif enrichment analysis was confounded by the close homology of bHLH binding sequences. Hence, to reinforce our confidence in the biological significance of these findings, we identified among the list of bHLH transcription factors those specifically up-regulated in the PP–DP transition as a proxy for their putative link with demethylation during neurogenesis. This led to the selection of only six factors, including Neurod1-2, Neurog1-2, Bhlh22, and Nhlh1 (Fig 3B), whose binding motifs ranked among the top-20 most enriched specifically for 5mC-loss/5hmC-gain, but not 5mC-gain/5hmC-loss, loci (compare Figs 3C and S3C). Interestingly, these six factors are known to play key roles in brain development with both Neurod1 and Neurog2 being reported to act as pioneer factors modifying the chromatin landscape during neurogenesis (Pataskar et al, 2016; Smith et al, 2016). In turn, other factors in our list were also shown to drive demethylation by association to Tet enzymes (Donaghey et al, 2018; Sardina et al, 2018).

To provide an independent assessment of the putative binding of at least some of these bHLH transcription factors within 5mC-loss/5hmC-gain loci, we again took advantage of publicly available ChIP-seq datasets. In mapping the reads obtained for two such transcription factors, Neurod2 and Neurog2 (Bayam et al, 2015; Velasco et al, 2017), within 5mC-loss/5hmC-gain loci as previously carried out for histones marks, we indeed observed a strong positive correlation that was absent when considering the converse

5mC-gain/5hmC-loss loci (Figs 3D and S3D). Conversely, we also observed that a decrease in 5mC and increase in 5hmC occurred at Neurod2 as well as Neurog2 binding sites (Fig 3E), further strengthening our previous results.

Finally, we sought to provide one example and experimental validation of the potential biological significance of our observations and the role of (hydroxy-)methylation in controlling gene expression of neurogenic genes during corticogenesis. To this aim, we first had to implement a method to site-specifically manipulate 5(h)mC in vivo.

### An improved method to manipulate 5(h)mC reveals the role of methylation within a novel regulatory element of *Dchs1*

To assess the biological relevance of our observations, we decided to locus-specifically manipulate 5(h)mC by in utero electroporation with an inactive (dead) dCas9 fused to the catalytic domain of Tet1 (dCas9-Tet1) (Fig 4A). To this aim, we generated an all-in-one vector by adding to the dCas9-Tet1 construct a GFP as reporter and gRNAs targeting the region of interest. Differently from two previous reports, we decided to only use two gRNAs rather than several (Liu et al, 2016) and avoid the bulky SunTag system to deliver Tet1 (Morita et al, 2016), thus decreasing potential off-targets and steric hindrance at once.

To test the efficacy and specificity of our system, we performed electroporation of E13.5 mouse embryos and FAC-sorted GFP+ cells 48 h later (Fig 4A). Different control vectors were tested including a dCas9 fused to a catalytically inactive dTet1 delivered together with gRNAs against either an unspecific sequence (dCas9-dTet1$^{LacZ}$) or the target locus (dCas9-dTet1$^{Target}$) with the latter control being important to assess potential effects of dCas9 occupancy irrespective of Tet1 activity. Next, we selected four loci within regions containing 4–9 CpG dinucleotides (28 in total) using as a main criterion their high level of methylation (to increase the sensitivity of our assessment) but not necessarily all the features emerging from our analysis as characteristic of 5mC-loss/5hmC-gain loci. Bisulfite amplicon sequencing on FAC-sorted GFP+ cells was then used to assess methylation levels when using any of the three vectors, that is, (i) inactive and unspecific dCas9-dTet1$^{LacZ}$, (ii) inactive but specific dCas9-dTet1$^{Target}$, or (iii) both active and specific dCas9-Tet1$^{Target}$.

This revealed that although the levels of 5(h)mC appeared by all means similar among the two control vectors when averaging cytosines within the target region together (Fig 4B), the use of the inactive but specific dCas9-dTet1$^{Target}$ was still sufficient to induce several significant, and in some instances inconsistent, changes at

---

**Figure 2. 5mC-loss/5hmC-gain loci characterize neurogenic genes up-regulated during fate commitment.**
**(A)** Volcano plots depicting the loss (orange) and gain (blue) in 5mC (left) or 5hmC (right) from PP to DP and DP to neurons (as indicated). Number of significant (>50% change, $P < 0.05$) differentially (hydroxy-)methylated loci (top) and nearest genes (parentheses) are indicated. **(B, C)** Heat maps (left) and violin plots (right) representing 5(h)mC levels within a ±2-kb region across differentially (hydroxy-)methylated loci in each cell type (colors, as above) calculated as log$_2$(ChIP/control) or trimmed mean of M values (TMM), respectively. In this analysis, 5hmC-gain and 5mC-loss loci identified in DP (4,757) or neurons (3,716), respectively, were first taken as a reference (top panels) and the levels of the converse cytosine modification (i.e., 5mC or 5hmC for B and C, respectively) measured within these very same loci (bottom) revealing the extensive correlation in 5(h)mC changes. **(D)** Whiskers–box plots of 5mC and 5hmC levels (left and right, respectively) calculated for individual cytosines (n = 48) after (oxidative) bisulfite amplicon sequencing of six among all 5mC-loss/5hmC-gain loci (one examples of which is depicted in Fig S2C). Data are plotted as fold change relative to PP. **(E, F)** DAVID-gene ontology term enrichment (top three terms) (E) and expression levels (FPKM) (F) of 5mC-loss and 5hmC-gain loci in the PP–DP and DP–neuron transitions (as indicated). Note the high enrichment, specificity, and consistency of the GO terms and the highly significant change in gene expression throughout the neurogenic lineage. Statistical test = edgeR-modified $t$ test (A), Wilcoxon rank sum test (B, C, D and F).

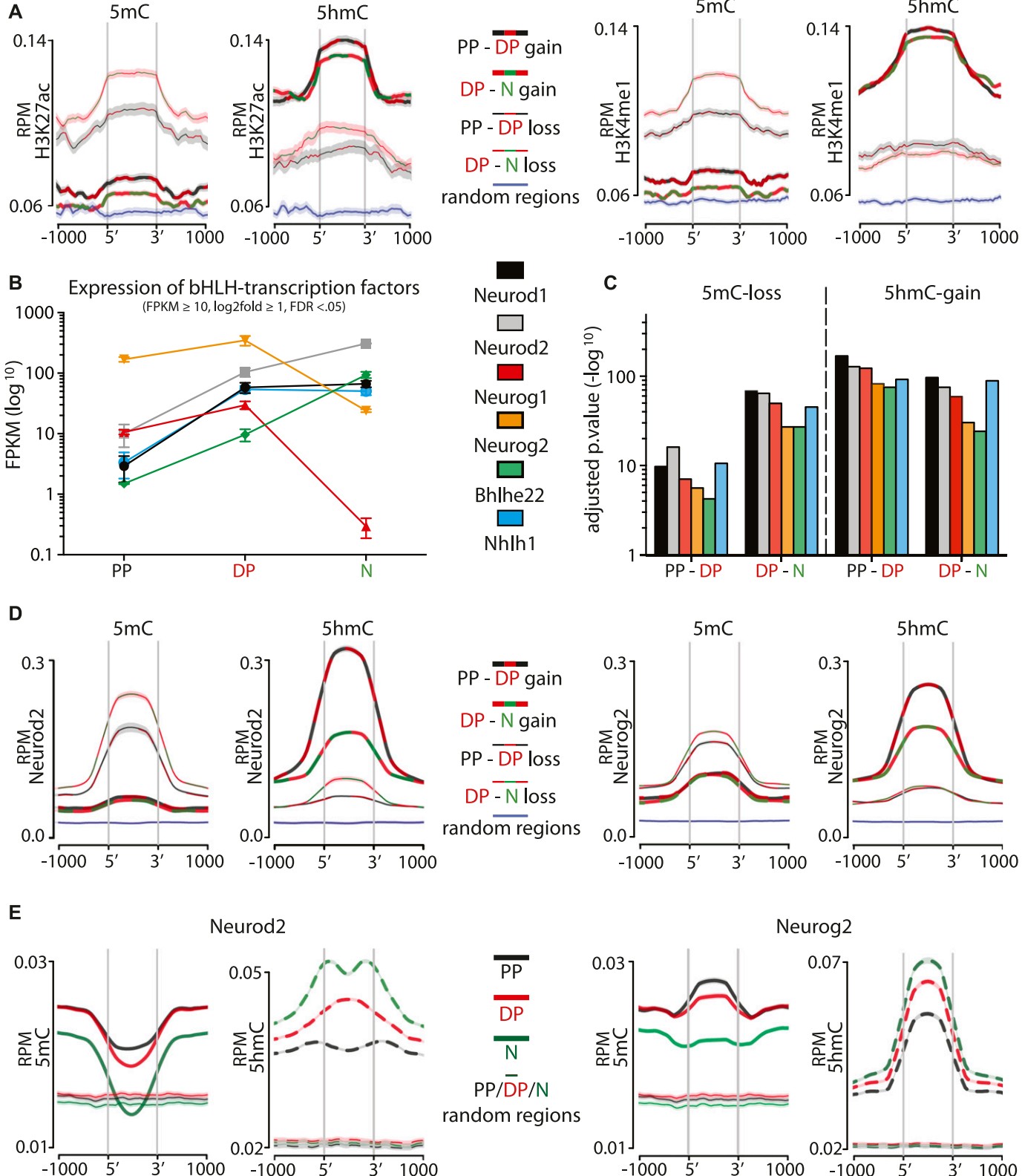

**Figure 3. 5mC-loss/5hmC-gain loci are enriched in active enhancers and pioneer factor–binding motifs.**
**(A)** Distribution and abundance (RPM) of active enhancer marks including H3K27ac (left) or H3K4me1 (right) as depicted across a ±1-kb region of loci undergoing differential (hydroxy-)methylation. For these plots, all eight possible combinations were considered, namely, gain (thick lines) or loss (thin lines) in 5mC (left) or 5hmC (right) in the PP–DP (black/red lines) or DP–neuron (red/green lines) transition as well as random loci as negative controls (blue lines). Note the increase in both

the level of individual cytosines (Fig S4A), which were likely due to dCas9 occupancy of the target locus irrespective of Tet1 activity. In contrast, only the active and specific dCas9-Tet1$^{Target}$ was found to induce a significant, major, and consistent reduction (~90%; $P < 0.001$) in methylation of the entire locus and across essentially all CpG dinucleotides (Figs 4B and S4A) but not affecting those within ~20 bp of the dCas9-binding region or outside (e.g., CpG 1 and 8 in Fig S4A, top left). While assessing the effects in (hydroxy-)methylation within the four target loci in embryos electroporated with active dCas9-Tet1$^{Target}$ constructs, we also assessed 5(h)mC of the remaining three loci finding no significant change relative to dCas9-dTet1$^{LacZ}$ controls (not shown) and validating both the efficiency and specificity of our approach.

We next verified the suitability of the previous control vectors to assess neurogenesis by comparing the distribution of electroporated cells in cortical layers 48 h after electroporation at E13.5 relative to an empty GFP control. Hence, we calculated the proportion of apical and basal progenitors and neurons by Tbr2 immunoreactivity and distribution of GFP+, electroporated cells across cortical layers (Tbr2– in the VZ, Tbr2+ in the VZ/SVZ, and Tbr2– in the IZ and CP, respectively). Unexpectedly, we found that electroporation with dCas9-dTet1$^{LacZ}$ had a minor, yet significant, increase in GFP+, apical progenitors and equivalent decrease in neurons, which was particularly noticeable in the CP relative to both GFP and active, but unspecific, dCas9-Tet1$^{LacZ}$ control vectors (Fig S4B). This was surprising because the dCas9-dTet1$^{LacZ}$ vector did not encode any functional protein, its effect on methylation was overall minor and inconsistent (Fig 4B), and a previous study using a similar vector did not report any negative effect on neurogenesis (Morita et al, 2016).

To gain more insight into this unexpected result, we repeated electroporation using only dTet1 vectors without dCas9 or gRNAs to exclude potential side effects of the former and off-targets of the latter. Once again, dTet1 alone triggered a subtle but significant change in the proportion of GFP+ cells, which was similar to the one previously observed using dCas9-dTet1$^{LacZ}$ constructs (Fig S4B). These subtle changes in any condition in which constructs encoding dTet1 were used resembled those previously reported in cell culture for the very same catalytically inactive enzyme (Gao et al, 2016) and suggested to result from dTet1 interaction with Gadd45a, a gene involved in neural development (Kienhofer et al, 2015). Therefore, considering the caveats arising from the use of dTet1 vectors, we concluded that dCas9$^{Target}$ was a superior negative control that, contrary to empty vectors such as GFP, still accounts for potential side effects due to dCas9 occupancy and hindrance of the target locus.

Having established our approach, we next sought to provide one example of the potential biological relevance of our observations

on differential (hydroxy-)methylation during neurogenic commitment by targeting a 5mC-loss/5hmC-gain locus. To this aim, we selected candidates among target regions that epitomized all the key features emerging from our study including: (i) a 5mC-loss/5hmC-gain in DP, (ii) within, or in proximity to, genes containing Neurod2 and Neurog2 binding motifs, and (iii) whose nearby gene was physiologically up-regulated during fate commitment. Among these, as expected from our previous gene ontology enrichment analysis (Fig 2E), we found numerous markers and characterized regulators of neurogenesis such as *Emx1*, *Prox1*, *Eomes*, *Pou4f1*, *Dll3*, *Tcf4*, *Tubb3*, *Wnt* members, and others. We finally selected *Dchs1* because of its relatively recent identification as a novel gene controlling the proliferation of neural progenitors and whose mutations is cause of developmental malformations, including heterotopia, in Van Maldergem syndrome (Cappello et al, 2013). Beyond this, and as one additional advantage of choosing *Dchs1* as a target for manipulation, its differential (hydroxy-)methylation occurred within a locus predicted, but not validated, to act as an enhancer that we sought to simultaneously assess by our experiments. In particular, this relatively tiny, 300-bp putative enhancer within an intron of a complex gene >35 kb in length and comprising 23 exons (Fig 4C, top) was of doubtful significance making it for a particularly challenging target to test our method and biological significance of our observations on (hydroxy-)methylation.

Hence, after confirming the differential (hydroxy-)methylation physiologically occurring within the *Dchs1* locus in PP, DP, and neurons by bisulfite sequencing (not shown), we designed dCas9-Tet1$^{Dchs1}$, or dCas9$^{Dchs1}$ as control constructs that were used to electroporate E13.5 mouse embryos. FAC-sorting of GFP+ cells 48 h later followed by bisulfite sequencing confirmed, also in this additional case, the efficacy of our approach leading to an overall ~50% decrease in 5(h)mC relative to control (Fig 4C). In these experiments, we additionally assessed potential off-targets of dCas9-Tet1$^{Dchs1}$ and dCas9$^{Dchs1}$ within the 10 genomic regions having the highest homology to the *Dchs1* gRNAs (5 each). Bisulfite amplicon sequencing of all these regions revealed no significant change in methylation at the level of all loci as well as individual cytosines (Fig 4D and data not shown), again highlighting the high specificity of our approach. Next, we investigated whether or not a loss in methylation alone was sufficient to change the expression of *Dchs1* by performing qRT–PCR on GFP+ FAC-sorted cells, which resulted in a 2.5-fold up-regulation of *Dchs1* relative to control (Fig 4E). Importantly, this result confirmed three assumptions at once: (i) it validated the regulatory function of the predicted enhancer within *Dchs1*, (ii) it showed that differential (hydroxy-)methylation within a 5mC-loss/5hmC-gain locus is a cause, rather than consequence, of gene up-regulation, and (iii) it showed that demethylation alone can potentially account for the observed up-regulation in

---

histone marks sharply at the 5′ and 3′ end of 5mC-loss/5hmC-gain, but not 5mC-gain/5hmC-loss, loci (heat maps are also shown in Fig S3A to appreciate individual loci). ChIP-Seq data for H3K27ac and H3K4me1 from the E14.5 mouse cortex were taken from the ENCODE project (Shen et al, 2012). Shadows represent the variance (SD) of the mean that in some cases was too small to be visibly depicted. **(B, C)** Expression levels (FPKM) of bHLH transcription factors up-regulated from PP to DP (B), and enrichment values of their binding motifs within 5mC-loss/5hmC-gain loci (C) (top five motifs are shown in Fig S3B). **(D)** Distribution and abundance (RPM) of Neurod2 and Neurog2 ChIP-Seq reads across a ±1-kb region including loci undergoing any of the eight possible combinations of differential (hydroxy-)methylation depicted as in A and including random loci as negative control (variance was too small to be depicted). **(E)** Converse analysis as in (D) with distribution and abundance (RPM) of 5(h)mC reads of PP (black), DP (red), N (green) plotted across Neurod2 (left–) and Neurog2 (right)–binding sites (bold) and random regions as negative control (left and right; thin lines).

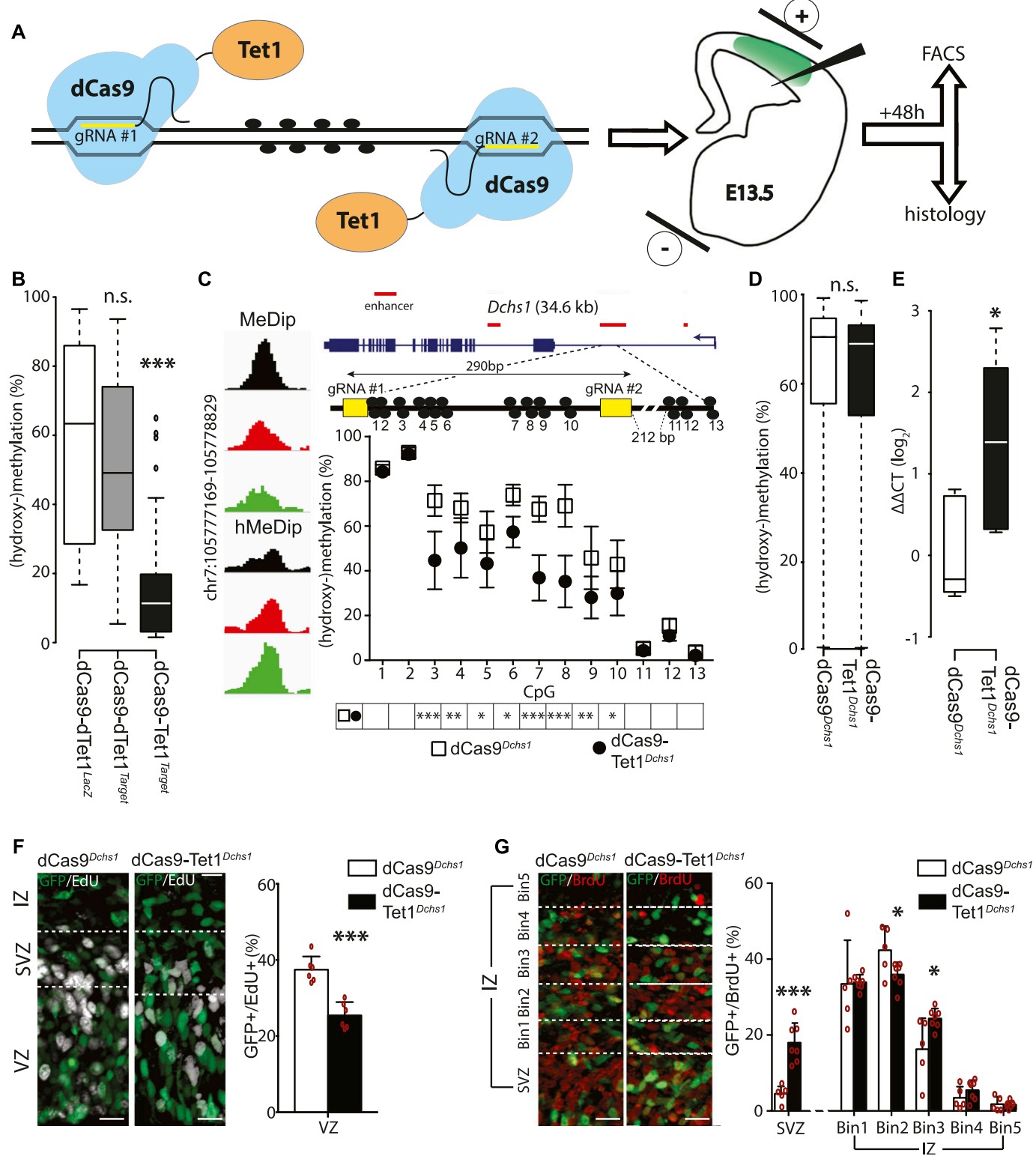

**Figure 4. Site-specific manipulation of (hydroxy-)methylation.**
**(A)** Drawing depicting dCas9-Tet1 targeting of a locus of interest through the use of two gRNAs (yellow) to manipulate 5(h)mC at CpG dinucleotides (black circles) by in utero electroporation (right) of a vector including all components of interest plus GFP as reporter. **(B)** Whiskers–box plots depicting 5(h)mC levels of FAC-sorted GFP+ cells 48 h after in utero electroporation as assessed by bisulfite sequencing of 48 cytosines within four loci (individual loci and values are shown in Fig S4A). Note the major reduction in 5(h)mC upon use of dCas9-Tet$^{Target}$ constructs. **(C)** Screenshot of 5(h)mC levels in cell types (left) and drawing (in scale) of the *Dchs1* locus and regions targeted by dCas9-Tet$^{Dchs1}$ vectors (top). Assessment of 5(h)mC in GFP+ cells 48 h after in utero electroporation was performed as described in Figs 4B and S4A

*Dchs1* during neurogenesis because the magnitude of the change induced by dCas9-Tet1$^{Dchs1}$ (2.5-fold) was nearly identical to that occurring from PP to neurons in physiological conditions (2.2-fold) (Aprea et al, 2013).

Moreover, as mentioned above, down-regulation of *Dchs1* was described to increase progenitor proliferation and alter neuronal migration (Cappello et al, 2013). Hence, we finally assessed whether a converse up-regulation of *Dchs1* by targeted demethylation would reach the levels that are necessary to trigger a biological effect consistent with this previous report. Indeed, immunohisto-chemistry 48 h upon electroporation at E13.5 with dCas9-Tet1$^{Dchs1}$ and EdU administration 6 h before sacrifice showed a 30% decrease (from 37.4 ± 3.5 to 25.4 ± 3.5%; *P* < 0.001) in proliferation and EdU+/GFP+ cells relative to dCas9$^{Dchs1}$, control vectors (Fig 4F). Moreover, BrdU birthdating 24 h before sacrifice revealed that the migration of newborn neurons was also impaired with a substantial, threefold increase (from 4.3 ± 2.3 to 17.7 ± 5.2%; *P* < 0.001) in the proportion of BrdU+/GFP+/Tbr2− (i.e., excluding basal progenitors) cells in the SVZ and their altered distribution in medial bins of the IZ, neuronal layer (Fig 4G).

Altogether, the observed decrease in proliferation of progenitors and altered neuronal migration induced by demethylation-driven up-regulation of *Dchs1* were consistent with a previous report (Cappello et al, 2013) and showed that demethylation within this newly identified regulatory region of *Dchs1* alone is both sufficient to trigger its up-regulation to the levels observed in physiological conditions and necessary to produce a biological effect.

## Discussion

Regulation of DNA (hydroxy-)methylation in stem cells and its role in controlling gene expression during fate commitment have been the focus of many studies resulting in conflicting hypotheses with regard to the causes underlying these changes (Calo & Wysocka, 2013; Smith & Meissner, 2013; Schubeler, 2015; Atlasi & Stunnenberg, 2017). Here, we provided the first resource describing the patterns of 5mC and 5hmC in specific cell types of the developing mammalian cortex. We further relate this to the gene expression program underlying specification of distinct cell fates during cortical development. We found that a switch from proliferative to neurogenic divisions was characterized by a gain in 5hmC in DP followed by a loss in 5mC in newborn neurons. These changes occurred predominantly within regions annotated as enhancers and enriched in pioneer transcription factor–binding motifs, which ultimately correlated with up-regulation of their nearby neurogenic genes. We then assessed the potential biological significance of our observations by developing an improved method to site-specifically manipulate 5(h)mC in vivo. By this, we identified a novel regulatory element within a factor involved in cortical

malformations: *Dchs1*. Several aspects of our study are worth discussing.

First, and contrary to our study, a previous description of the (hydroxy-)methylome of mixed populations of progenitors versus neurons did not observe a loss in 5mC during neurogenesis (Hahn et al, 2013). This inconsistency is likely explained not only by our discrimination of PP and DP as distinct cell populations but also by the higher resolution of our sequencing analysis relative to the microarrays previously used. In addition, Hahn et al (2013) used MIRA as a method to detect 5mC, which strongly favors its identification within CpG islands despite the fact that our and previous studies (Guo et al, 2011; Schubeler, 2015; Yao et al, 2016) have shown that CpG islands are not target of differential (hydroxy-)methylation. As a result, by also considering DP as an intermediate cell type with a bivalent epigenetic signature, our study reveals the full dynamics of cytosine modifications that may be critical to decipher the role of this epigenetic mark in corticogenesis.

Second, we found that a gain in 5hmC is initiated in DP resulting in a loss in 5mC in their neuronal progeny. These 5mC-loss/5hmC-gain loci (i) occurred within, or in close proximity to, neurogenic genes, (ii) were enriched in binding motifs of bHLH, among them pioneer, transcription factors, and (iii) their change in (hydroxy-)methylation correlated with an activation of enhancers and up-regulation of gene expression. Notably, these genes included many known regulators and markers of corticogenesis, such as *Emx1*, *Prox1*, *Eomes*, *Dll3*, *Tcf4*, *Tubb3*, *Wnt* family members, and *Dchs1*, to mention a few, which in essentially all cases were up-regulated in the PP–DP–neuron lineage. In describing the features of 5mC-loss/5hmC-gain loci, we also observed that the converse pair of cytosine modifications, namely, 5mC-gain/5hmC-loss, consistently displayed unrelated characteristics such as a diverse enrichment in genomic features, transcription factor–binding motifs, and so forth. This does not imply that these latter modifications are irrelevant in neurogenesis as in fact, several sites among this group were adjacent to positive regulators of stemness and pluripotency. For example, a gain in 5mC was observed at loci for *Nanog*, *Shh*, *Numb*, *Pou3f2*, and others and whose expression decreased in the PP-DP transition consistent with a functional role during differentiation. In addition, 5mC-gain loci showed an enrichment in binding motifs for methylation-sensitive transcription factors and negative regulators of neurogenesis such as Hif1a and Hes1 (Koslowski et al, 2011; Yin et al, 2017). As such, our data highlight that a gain in 5mC and loss in 5hmC, although still important, are uncoupled from each other and controlled independently through mechanisms yet to be identified.

Third, in addition to providing an improved method for dCas9-mediated manipulation of (hydroxy-)methylation, we assessed the significance of our observations by validating the causal link

(bottom). **(D)** Whiskers–box plots depicting 5(h)mC levels assessed by bisulfite sequencing at top 10 predicted off-target regions (5 for each gRNA, 51 CpGs total) of FAC-sorted GFP+ cells 48 h after in utero electroporation of dCas9-Tet$^{Dchs1}$ or dCas9$^{Dchs1}$. **(E)** *Dchs1* expression levels (ΔΔCT) calculated by qRT–PCR on GFP+ cells as described in (C). Note the ~2.5-fold increase expression upon demethylation. **(F, G)** Pictures of the E15.5 mouse cortex (left) and quantifications (right) 24 h after electroporation under different conditions followed by (immuno-)labeling for GFP, EdU, or BrdU (as indicated). Boundaries of the SVZ with the VZ and IZ were assessed by Tbr2 staining (not shown) or equidistant bins through the IZ are indicated (dashed lines in E and F, respectively) in which GFP+ cells were expressed as proportion of their total number. Scale bar = 50 *μ*m; bars = SD (SEM in C); n ≥ 6 (E and F); statistical test = *t* test; *P < 0.05; **P < 0.01; ***P < 0.001.

between demethylation, up-regulation of gene expression, and regulation of corticogenesis for one paradigmatic gene containing a 5mC-loss/5hmC-gain locus: *Dchs1*. This gene has recently emerged as a novel regulator of brain development whose mutations cause Van Maldergem syndrome, an autosomal-recessive disorder characterized by intellectual disability and skeletal malformations (Cappello et al, 2013). In fact, knockdown of Dchs1 in the developing cortex was shown to cause heterotopia as a result of an increased cell proliferation and delayed neurogenesis (Cappello et al, 2013), which is consistent with our converse manipulation leading to increased expression of *Dchs1*, reduced proliferation and altered neuronal migration. Although the previously identified *Dchs1* mutations causing heterotopia result in a truncated protein (Cappello et al, 2013), it is here intriguing to consider the possibility that other developmental malformations may involve cytosine residues target of (hydroxy-)methylation and resulting in aberrant gene expression. Such a role of DNA modifications in disease is supported by the observation that methylated cytosines are hotspots for single-nucleotide polymorphisms (Schubeler, 2015) and changing the methylation of even one single cytosine residue may result in changes in expression levels (Furst et al, 2012). In this context, it is worth mentioning that our manipulation of *Dchs1* led to a reduction by roughly half in methylation of just eight CpG dinucleotides within a small region of a relatively big, multiexonic gene. It is thus remarkable that this alone was sufficient to double the expression of *Dchs1* to levels similar to those observed during physiological differentiation. Therefore, the recent possibility to recapitulate (hydroxy-)methylation patterns in human iPS-derived organoids (Luo et al, 2016) combined with their site-specific manipulation opens up a new dimension in studying the role of epigenetics in diseases of the brain and other organs.

Finally, fourth, our data are in line with recent observations pointing out that the binding of pioneer transcription factors per se can cause demethylation (Donaghey et al, 2018; Sardina et al, 2018). In fact, at least for the transcription factors identified in our study, their expression was specifically increased in DP and, hence, became more abundant in the appropriate cell type and at the appropriate time for initiating hydroxymethylation. In addition, although DNA binding of transcription factors is generally inhibited by methylation, these bHLH pioneer factors are methylation insensitive (Yin et al, 2017) and, thus, display the appropriate feature needed to initiate and drive this process. In light of this, a model has been recently reviewed in which pioneer transcription factors drive the activation of enhancers (Atlasi & Stunnenberg, 2017). Our data are consistent with this model and would even extend it in the context of DNA demethylation which, together with changes in histone marks and opening of the chromatin, concur toward the activation of enhancers, up-regulation of nearby neurogenic genes and, hence, cell fate change (Fig 5). The bivalent (hydroxy-)methylation footprint of DP is also fully consistent with this model and suggests a dual function to first, fate-commit DP to switch from proliferative to neurogenic division and second, to consolidate neuronal identity in the resulting neurons. This is reminiscent and would actually explain the reported onset of expression of neuron-specific genes already at the level of DP (Aprea et al, 2013) as if this cell type was *primed* to acquire the early features of their future neuronal progeny. Clearly, key aspects of this model are still highly debated such as whether DNA methylation is a cause or a consequence of gene up-regulation and cell fate change (Stricker & Götz, 2018). At least for Dchs1, our study shows that demethylation alone can be a driver of these processes.

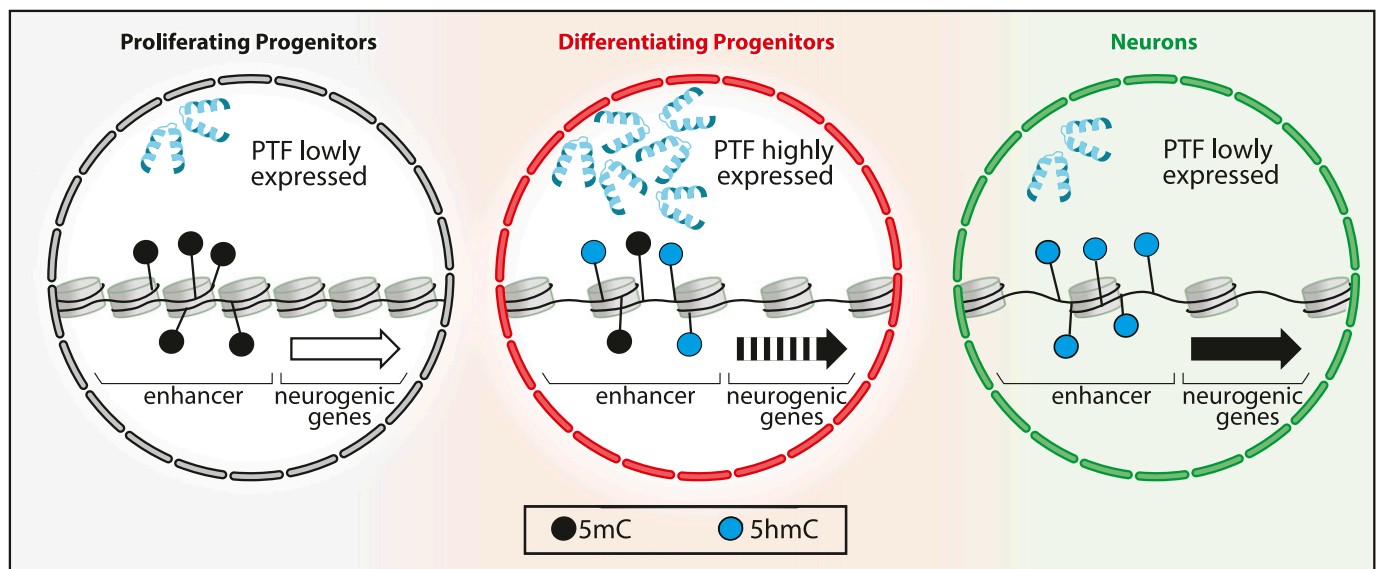

**Figure 5.  DNA (hydroxy-)methylation in neurogenic commitment.**
A model emerging from several studies (see text) suggests that pioneer transcription factors (PTFs) drive cell fate commitment and differentiation by remodeling chromatin. In this context, our data suggest that PTF up-regulated during the PP–DP transition are associated with oxidation of 5mC to 5hmC (black to blue dots) within enhancers of neurogenic genes. This results in priming of those genes (dotted to continuous arrow) and fate commitment of DP toward neurogenic division. This process is continued during the next cell division (DP-N) leading to higher levels of 5hmC, reinforced gene expression (continuous to bold arrow), and establishment of neuronal identity despite a possible down-regulation of PTF.

# Materials and Methods

## Animals and cell sorting

Mice were kept under standard housing conditions, and experiments were carried out according to local regulations. E14.5 *Btg2*[RFP]/*Tubb3*[GFP] double heterozygous embryos or E13.5 C57bl/6J pregnant mice (Janvier Laboratories) were used for (h)MeDIP or in utero electroporation, respectively. For the latter, the mice were treated as previously described (Aprea et al, 2013) by injecting ~3 µg of endotoxin-free DNA into the telencephalic ventricles followed by delivery of eight 50-ms-long electric pulses of 36 V with intervals of 1 s. The mice were euthanized 48 h later eventually upon intraperitoneal administration of 1 mg BrdU (1 pulse at E14.5) and/or 0.1 mg EdU (3 and 6 h before sacrifice). Brains were either dissected and processed for immunohistochemistry or dissociated by the papain-based neural dissociation kit (Milteney Biotec). FAC-sorting was performed at 4°C in the four-way purity mode with a flow rate of 20 µl/min and side and forward scatter light to eliminate debris and aggregates gating for green (488 nm) and red (561 nm) fluorescence as described (Aprea et al, 2013). The cells were sorted into DNA/RNA lysis buffer (QIAamp DNA Micro, QIAGEN or Quick-RNA MicroPrep Zymo Research) or pelleted (500 g, 5 min at 4°C) for later use.

## (h)MeDIP and sequencing

Comparative 5(h)mC immunoprecipitation ((h)MeDIP) was adapted from Tan et al (2013). Briefly, 1 µg of DNA from PP, DP, or neurons in three independent biological replicates was sheared using the Covaris LE220 (duty cycle 10%, intensity 5, cycles/burst 200, 180 s) into ~250 bp fragments that were ligated to barcoded sequencing adaptors (NEBNext Ultra DNA Library Prep kit for Illumina; New England Biolabs). 300 ng of DNA from each cell type was pooled and (h)MeDIP performed using the MagMeDIP or hMeDIP kits (Diagnode) according to the manufacturer instructions. The immunoprecipitated DNA and non-antibody–treated control was quantified, amplified (15 cycles; NEBNext High-Fidelity 2X PCR Master Mix; New England Biolabs), gel size selected (225–325 bp) using the E-Gel EX system (2%) (Invitrogen), and used for 75-bp single-read sequencing on an Illumina HiSeq 2000 platform resulting in ~30–60 million reads per sample. Sequencing raw data were deposited in Gene Expression Omnibus (GEO) database: GSE104585.

## Bioinformatics and statistics

Raw sequencing reads were aligned to the mm[10] reference genome using BWA v0.7.8 and peaks called using SICER v1.1 (window = 25 bp, gap = 50 bp, fragment = 250 bp, max redundant reads = 2, and FDR = 0.0001). Peaks found in all replicates of each cell type were merged to build a reference peak sets of PP, DP, and neurons for 5mC and 5hmC excluding those overlapping repetitive sequences (RepeatMasker). Diffbind v2.6 was used to calculate TMM (DBA_SCORE_TMM_READS_EFFECTIVE) values and identify differential (hydroxyl-)methylated regions (DBA_edgeR, 50% change, $P < 0.05$) that were annotated using HOMER v4.7. Genomic features, GO terms, or enrichment of transcription factor–binding motifs were assessed by HOMER v4.7, DAVID v6.8, or MEME-AME v5.0 (JASPAR CORE and UniPROBE mouse), respectively. ChIP-seq data from the ENCODE or GEO databases were used to generate bigwig files calculating the $\log_2$(control/ChIP) using deepTools v3.0.2 (bin size = 50, scaling method = SES) that was also used together with NGSplot v2.61 (robust statistics fraction = 0.05) to generate profile plots or heat maps, respectively. Off-targets of gRNAs were predicted using Cas-OFFinder (http://www.rgenome.net/cas-offinder/). Z-scores were calculated using regioneR (1.14.0) permutation test (n = 1,000). Bar and whiskers–box plots were built using RStudio v1.0.143 and *t* test, or Wilcoxon rank sum test was used to assess significance as appropriate.

## Constructs

The dCas9-Tet1 fusion protein with a 9–amino acid linker (LQGGGSGS) was generated by replacing the DNMT3a domain within pdCas9-DNMT3A-EGFP (Vojta et al, 2016) with the active or inactive catalytic domain of human TET1 (Maeder et al, 2013) with cassettes being introduced (Table S1) for one (LacZ) or two gRNAs under independent U6 promoters. The (d)Tet1 cassette was removed by enzymatic digestion (FseI/XhoI) to obtain the dCas9 control plasmid.

## Immunohistochemistry

Immunohistochemistry was performed as described (Aprea et al, 2013). Briefly, the brains were fixed in 4% paraformaldehyde in 0.1 M phosphate buffer (pH 7.4; PFA) at 4°C for 12 hours, cryoprotected in 30% sucrose, and cryosectioned (10 µm). Sodium citrate (0.01 M; pH 6.0)–based antigen treatment was performed (1 h at 70°C) followed by incubation with the appropriate antibodies (Table S2), washes with PBS, and PFA post-fixation for 30 min. For EdU, the Click-iT Alexa Fluor 647 kit was used according to the manufacturer's instructions (Invitrogen). For BrdU, the sections were incubated in 2 M HCl after post-fixation for 30 min at 37°C, followed by washes with PBS and antibody incubation.

## (Oxidative-)bisulfite amplicon sequencing and qRT–PCR

(ox)BSamp-seq protocol was adapted from Masser et al (2015). Briefly, 0.2–1 µg of DNA was converted either using the EpiTect Bisulfite kit (QIAGEN) for BSamp-seq or the TrueMethyl-seq kit (Cambridge Epigenetix) for oxBSamp-seq. Converted DNA was used as a template to PCR-amplify the region of interest using bisulfite-specific primer pairs (Table S1). The purity and size of the amplicons was validated on gels and purified using the QIAquick PCR Purification Kit (QIAGEN). 0.2 µM from all amplicons of each sample were pooled and used for tagmentation-based whole-genome library preparation using the Nextera DNA Library Preparation Kit (Illumina). Libraries were amplified using the KAPA High Fidelity Master Mix (PeqLab), purified using Ampure XP-beads (Beckman & Coulter), and loaded on a Miseq-Nanoflowcell (Illumina). Each sequencing run resulted in about 100,000 reads distributed over all pooled amplicons. Data were analyzed using Bismark 0.16.0. For qRT–PCRs, reverse transcription was performed with 30–100 ng of RNA using

the iScript cDNA Synthesis kit (Bio-Rad). Transcripts were quantified by iQ SYBR Green Supermix (Bio-Rad) on a CFX Connect real-time PCR Detection System (Bio-Rad) using the appropriate primers (Table S1).

## Supplementary Information

## Acknowledgements

This work was supported by the Center for Regenerative Therapies TU Dresden (CRTD), the German Research Foundation (SFB655 project A20) and a fellowship of the DIGS-BB of the TU Dresden awarded to F Noack. We are grateful to Andreas Dahl, Annekathrin Kränkel, and Susanne Reinhardt of the Dresden Genome Sequencing Center for helpful discussions and the MPI-CBG and CRTD animal house for support. Animal experiments were approved by local authorities (24-9168.11-1/41 and TVV 39/2015).

### Author Contributions

F Noack: conceptualization, data curation, formal analysis, validation, visualization, methodology, project administration, and writing—original draft, review, and editing.
A Pataskar: data curation.
M Schneider: methodology.
F Buchholz: conceptualization and resources.
VK Tiwari: conceptualization, formal analysis, and writing—review and editing.
F Calegari: conceptualization, supervision, funding acquisition, project administration, and writing—original draft, review, and editing.

### Conflict of Interest Statement

The authors declare that they have no conflict of interest.

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
