## [Reviewer comments · Life Science Alliance]

Life Science Alliance

Assessment and Site-Specific Manipulation of DNA (Hydroxy-)Methylation During Mouse Corticogenesis

Florian Noack, Abhijeet Pataskar, Martin Schneider, Frank Buchholz, Vijay Tiwari, and Federico Calegari

DOI: <https://doi.org/10.26508/lsa.201900331>

Corresponding author(s): Federico Calegari, DFG-Research Center and Cluster of Excellence for Regenerative Therapies, Faculty of Medicine, Technische Universität Dresden

Review Timeline:	Submission Date:	2019-02-04
	Editorial Decision:	2019-02-05
	Revision Received:	2019-02-18
	Editorial Decision:	2019-02-18
	Revision Received:	2019-02-19
	Accepted:	2019-02-20

Scientific Editor: Andrea Leibfried

Transaction Report:

Please note that the manuscript was previously reviewed at another journal and the reports were taken into account in the decision-making process at Life Science Alliance. Since the original reviews are not subject to Life Science Alliance's transparent review process policy, the reports cannot be published.

February 5, 2019

Re: Life Science Alliance manuscript #LSA-2019-00331-T

Prof. Federico Calegari
DFG-Research Center and Cluster of Excellence for Regenerative Therapies, Faculty of Medicine,
Technische Universität Dresden
Fetscherstr. 105
dresden, Dresden 01307
Germany

Dear Dr. Calegari,

Thank you for transferring your manuscript entitled "Assessment and Locus Specific Manipulation of DNA (Hydroxy-)Methylation Reveals the Dynamics of Cytosine Modifications During Neurogenic Commitment" to Life Science Alliance. The manuscript was assessed by expert reviewers at another journal before, and the editors transferred those reports to us with your permission.

The reviewers thought that your work is robust, but lacks further reaching molecular insight such as elucidating the involvement of TET proteins in the observed alterations of DNA (Hydroxy-)methylation status. They further thought that alternative interpretations of the data are possible and that further validation experiments need to be shown. The lack of in-depth molecular insight is not a concern for publication here, and we would thus like to invite you to submit a revised version to us. We would expect a point-by-point response to the reviewer concerns and accordingly:

Reviewer #1: text changes to address points 1-3, 6, 7. Addressing point 4 and point 8. Point 5 does not need to get addressed.

Reviewer #2: text changes to address main concerns; addressing 'additional comments' with the data already at hand.

Reviewer #3: address the criticism regarding validation experiments; resolve the criticism regarding 'bivalency', respond to 'additional points'.

Thank you for this interesting contribution to Life Science Alliance. We are looking forward to receiving your revised manuscript.

Sincerely,

Andrea Leibfried, PhD
Executive Editor

Life Science Alliance
Meyerhofstr. 1
69117 Heidelberg, Germany
t +49 6221 8891 502
e a.leibfried@life-science-alliance.org
www.life-science-alliance.org

- A letter addressing the reviewers' comments point by point.
- An editable version of the final text (.DOC or .DOCX) is needed for copyediting (no PDFs).
- High-resolution figure, supplementary figure and video files uploaded as individual files: See our detailed guidelines for preparing your production-ready images, <http://life-science-alliance.org/authorguide>
- Summary blurb (enter in submission system): A short text summarizing in a single sentence the study (max. 200 characters including spaces). This text is used in conjunction with the titles of papers, hence should be informative and complementary to the title and running title. It should describe the context and significance of the findings for a general readership; it should be written in the present tense and refer to the work in the third person. Author names should not be mentioned.

B. MANUSCRIPT ORGANIZATION AND FORMATTING:

Full guidelines are available on our Instructions for Authors page, <http://life-science-alliance.org/authorguide>

REVIEWER 1

We thank the reviewer for finding our work interesting and hope to have addressed the remaining criticism in the following point-by-point response and revised manuscript.

Author response to point 1 of this reviewer:

The role of Tets and their functional inactivation are described in the literature (He et al., 2011; Hahn et al., 2013 and others). Consistently, evidence for the oxidase-mediated pathway in our study comes from the observation that 5hmC increases at the same loci that subsequently lose 5mC (Fig. 2B and C). The reviewer is correct that experimental validation of Tet binding would be valuable. We made several attempts in the past by Tet1-ChIP-Seq. However, we repeatedly failed due to limited amount of material and suboptimal antibodies. Not least, it is unknown which Tet (1-2-3) is more relevant in our context and, hence, we would have to repeat these experiments 3 times. This goes well beyond our resources and agree with the editors at Life Science Alliance that these experiments are not essential for the current study.

Author response to point 2 of this reviewer:

That pioneer transcription factors per se can cause demethylation has been shown by other groups (Suzuki et al., 2017; Donaghey et al., 2018; Sardina et al., 2018). In light of this, it is correct that our data are consistent, hence support the notion, proposed by these and other authors (reviewd by Atlasi et al., 2017). We now tried to make it very clear that we only interpreted our data in the frame of the literature and models proposed by others, not us.

Author response to point 3 of this reviewer:

In genome-wide studies, loci are conventionally classified as “enhancers” when carrying enhancer-associated marks such as H3K27ac, H3K4me1 and open chromatin state. This was assessed in tissues from many species by international consortia (ENCODE and Ensembl) using ChIP-seq, ATAC-seq and/or DNase hypersensistivity assays. The reviewer asks if any of these loci were “truly characterized as enhancers”. This would require the generation of enhancer-associated, transgenic, reporter lines, which has not been done but for a negligible number of such loci. We now specify the origin of these annotations throughout the manuscript.

Author response to point 4 of this reviewer:

The reviewer is correct that CpG islands are depleted in 5mC when referring to the 5′end of active promoters. Conversely, high 5mC was found at inter-genic and in particular intra-genic CpG islands specifically in the brain (Maunakea et al., 2010). The reason why Fig. 1C appears inconsistent with the literature is that here all, inter-, promoter- and intra-genic regions were merged together as an average. Indeed, splitting these data in the 3 components (left panel below) reflects the known patterns. A similar effect also explains the overall levels of 5mC at enhancers with the vast majority showing low levels except for a small group with extremely high levels (heat maps, below). This clearly influences the final average values in Fig. 1C.

We appreciate that adding these data in the original Fig. 1C may be valuable but find it difficult to do so in this already crowded figure without compromising clarity of the overall message (CpG islands are not the focus of our study and enhancers are analyzed in greater details in the following figures). We therefore prepared these data only for the reviewer, that will become available according to the Journal's transparent process. We also remind that all our data will be freely available and that any reader will be able to inspect any given region of his/her choice.

Author response to point 5 of this reviewer:

These are all excellent questions and some of them already addressed in the literature (e.g. Suzuki et al., 2017; Donaghey et al., 2018; Sardina et al., 2018). We hope that the reviewer would agree that addressing this further will go beyond the purpose of the current study and, regrettably, our resources. Following consultation with the editors at Life Science Alliance, further experiments in this direction were not deemed essential for publication.

Author response to point 6 of this reviewer regarding negative control in Figure 4:

We thank the reviewer for this comment and reply breaking it up in its 3 parts.

i) A negative control for demethylation should, by definition, not trigger any demethylation of any locus. This excludes the use of active Tet1. Of the remaining two options, one could either use a) the inactive dTet1 or b) any empty GFP, dCas9 or other vector. With regard to the first option, electroporation of any dTet1 vector (even alone without dCas9 nor gRNAs!) induced minor but significant changes in cell distribution across layers relative to well established GFP controls. This subtle effect by dTet1 was also reported by other groups (Gao et al., 2016; Kienhofer et al., 2015), which we find important to corroborate for labs starting similar approaches. After excluding both Tet1 and dTet1, empty vectors were considered. GFP is a standard negative control used by dozens of labs but this would not account for potential effects triggered by dCas9-occupancy of the target locus that, indeed, we found to influence methylation (Fig. S4). Hence, we concluded that dCas9 is the best negative control.

ii) It is unclear to us why the reviewer says that the region in *Dchs1* does not contain transcription factor binding motifs or enhancer marks. Below we included a screenshot of the relevant locus showing differential methylation and containing both transcription factor motifs and marks of open chromatin (red rectangle and surrounding regions).

iii) Finally, the reviewer is entirely correct that the 4 regions chosen in Fig. S4 are neither related to the rest of our study nor showing all the features that we describe. In fact, they were not meant to show them. Our study combines a resource and a new method and these 4 regions were selected solely for the purpose of establishing the latter, not validating the former. As the main requirement for this, these regions were chosen as highly methylated making the assessment of the induced de-methylation more sensitive. Yet, it was not necessary for the establishment of our method that these regions had all the other features that applied to Dchs1, We now clarified this in our revised manuscript, see red text in page 13.

Author response to point 7 of this reviewer (figure 5):

We agree that demethylation must involve Tets and the known 5hmC-5fC-5caC-5C pathway. In fact, we consider this pathway so well described in many studies (e.g. Wu and Zhang 2011) that we did not find it particularly advantageous to repeat this validation again, even more so in the light of the complex biochemical analyses that would be necessary. In agreement with this, the editors at Life Science Alliance did not consider this validation essential.

Author response to point 8 of this reviewer:

In Fig. 1/S1 we start our analysis by merging all cell types together. Hence, these figures cannot show changes in different cell types. Comparison of different cell types is the focus of Fig. 2/S2 in which we now added new examples as requested by this and Reviewer 3. Clearly, more examples, comparisons and analyses can always be added. About this, we would like to remind that our raw data and processed files will be freely available as supplemental files and deposited in GEO. From this, anybody will be able to analyze our data for any locus and by any means.

REVIEWER 2

We thank the reviewer for appreciating the value of our work and hope to have addressed the remaining criticism in the following point-by-point responses and revised manuscript.

Author response to point 1 of this reviewer (lack of temporal resolution):

We agree that analyses across developmental time would be valuable. However, such analysis would be confounded by the appearance of additional cell types that cannot be resolved by our

reporter mouse. In addition, these experiments would go beyond a timely revision and for this reason not deemed essential by the editors at Life Science Alliance.

Author response to point 2 of this reviewer (heatmaps):

We are sorry for our lack of clarity. This is a critical point and we will dissect it in all details. The reviewer states that the patterns of the heat maps are “very different”. Indeed they are! They have “opposite patterns” because they inversely correlate. In essence, whenever loci are selected in which **5hmC increases** (Fig. 2B, top), here we found that **5mC decreases** (Fig. 2B, bottom). Vice versa, where **5mC decreases**, **5hmC increases** (Fig. 2C). Hence, in most cases, **5hmC/5mC negatively correlate**. This correlation specifically in the PP-DP and DP-N transitions is far from trivial because other combinations (5mC/5hmC, gain/loss, and PP/DN) did not show correlations, neither negative nor positive (e.g. Fig. S2A-B). This is a major point and novelty of our study showing that 5hmC-gain/5mC-loss loci are “functionally linked”.

Having said that, we agree that this negative correlation in the heat maps might be difficult to appreciate. This is because the same color-scale is used for both modifications (red/blue=0/10 TMM) while each starts with a different number of reads due to biological differences and use of antibodies with different avidity. We wish to make an example of this. Lets take 2B-left: Here the top 3 panels switch from “red” to “light blue” to “dark blue”, indeed **5hmC increases**. Conversely, the bottom 3 switch from “light blue” to “yellow” to “red” indicating that **5mC decreases**, clearly an inverse correlation. Yet the “light blue” does not match the middle “yellow” and the “dark blue” on top is absent in the bottom. From there, the eye may wrongly conclude that there is no correlation. This could easily be avoided by changing the scale of the bottom to, say, 0/5 rather than 0/10 TMM making the colors fit quite well. However, to our opinion this would not be appropriate for the sole purpose of tricking the eye and we find it important for readers to appreciate that the absolute abundance of the two modifications is different, which requires the same scale. Since quantifications and statistics are provided in the violin plots to the right, we finally decided to leave these figures as they are.

Author response to this reviewer regarding 5hmC potentially being a stable epigenetic mark:

The reviewer is correct. We believe that for most sites the 5hmC modification is permanent. With “intermediate” we did not mean “intermediate modification” but rather refer to DP as an “intermediate cell type”. We corrected this throughout the manuscript, including Fig. 5.

Author response to point 3 of this reviewer regarding Dchs1 enhancer demethylation as single functional assay:

Manipulation of this region was meant to provide just one example of the potential relevance of our observations. Indeed, de-methylation of this region triggered a phenotype and the same could apply to many other loci, although clearly not all to explain genome-wide changes. We revised the manuscript at several steps to make this overwhelmingly clear that we consider this just “one example” of the “potential significance” (e.g. page 13 and 15).

Author response to this reviewer regarding correlation between DNA (hydroxy)methylation and gene expression (Fig 2F):

This figure shows in fact gene expression associated with each modification in both PP-DP and DP-N (as requested, three cell types). Certainly, other associations could be assessed between our and other studies. About this, our raw data and processed files will become freely available and deposited in GEO. This also applied to our works on mRNAs (Aprea et al., 2013), lncRNAs (Aprea et al., 2015), Tox binding (Artegiani et al., 2013) and others on miRNAs and circRNAs to appear, hopefully, soon.

Author response to point of this reviewer regarding lack of experimental data related to pioneer activity:

As indicated above in response to Reviewer 1 point 2 and 5 the functional link between pioneer factor binding and methylation has been shown by other groups, not us (Suzuki et al., 2017; Donaghey et al., 2018; Sardina et al., 2018). Also, we are not aware of any method to measure “pioneer factor activity”, site-specifically, in vivo. As a proxy for “activity”, we took advantage of previous studies on pioneer factors “binding” in neural systems very similar, if not identical, to ours (Bayam et al., 2015; Velasco et al., 2017). The same we did for identifying histone modifications and enhancers as annotated in Ensembl and ENCODE.

Author response to point of this reviewer regarding “bivalent” 5mC/5hmC mark.

Yes, they do. The bivalent signature of DP can be seen in Fig. 1D, 2B-C. Note that in all cases, red indicators of DP are “in between” more extreme values of the other two cell types.

Author response to point of this reviewer regarding expression levels of TET proteins (Tet1/2/3) in PP, DP and neurons:

Disappointingly constant... we also hoped for changes but this was not the case (the reviewer can check any gene in the supplemental files from Aprea et al., 2013 and 2015). In turn, this suggests the involvement of co-factors, including pioneer factors, as discussed in our study.

Author response to point of this reviewer regarding Figure 1A and 1B:

Fig. 1A is a drawing without data, so we deduce that the reviewer refers to 1B (and perhaps 1C). We provide this below but this didn't show any clear difference that we could interpret. Since merging cell types did not result in any significant loss of information, we decided not to include these new plots while mentioning them as data not shown in our revised manuscript (page 7). The same applies to most other features in 1C (see also reply to Reviewer 1, point 4), except for enhancers that are indeed split for cell types in 1D.

Author response to point of this reviewer regarding Figure 2A, 2E/F, Figure 3:

These three points can be addressed at once as they all deal with the methodology intrinsic in bioinformatic analyses. Merging sets of high-throughput data to calculate “overlap” is usually not informative because any potential correlation, overlap or feature in dealing with thousands of values can, by definition, only apply to a subset of them (as the reviewer says: “a fraction”). The key question is whether or not this fraction is significantly over-represented (enriched) as expected from a random subset taken as a control. For example, when merging the 4,757 and 3,716 sites, as requested by the reviewer, it appears that only about 10% (487) of the former are also in the latter and leading to the wrong conclusion that the two do not correlate. Yet, the probability of this to happen by chance is astronomically low, meaning that the correlation is enormously strong. The reviewer can have a glimpse of the significance of such enrichments in Fig. 2B-C depicting p values at times reaching e^{-20} , the calculation limit set by the software!

On top of this, “merging groups” ultimately depends on arbitrary thresholds used to define both magnitude and significance to belong to those groups. While for practical reasons the numbers given in Fig. 2D were based on the same thresholds of magnitude and significance, there is no reason to assume that biologically functional changes in 5mC and/or 5hmC must have the same magnitude. Neither it is trivial to assume that the biological and/or technical variance of both groups should be the same, which in turn influences the significance. This is one additional reason why simply merging groups may not be particularly informative and why correlations in our study are better assessed by assessing the significance of enrichment.

Author response to point of this reviewer regarding Figure 3:

We also address these points at once. All these analyses are clearly possible. Indeed, predicted enhancers show a high enrichment in H3K4me1 as well as H3K27ac ChIP-seq signal from the E14.5 cortex and are at least partly bound by NeuroD2 and NeuroG2 (see panel A below). Neurod2/Neurog2 binding sites show a strong signal of the corresponding ChIP as well as H3K4me1 and H3K27ac (B). Indeed, all this can also be visualized as heatmaps (C). the reviewer can see all these analyses in the panels below. We also would like to remind that all

our data will be freely available and, hence, anybody will be able to re-analyze any set of data and depict them in any possible way.

Author response to point of this reviewer regarding Figure 4:

This is unfortunately not possible. Bisulfite and oxidative bisulfite reactions require very different amount of material, down to single cells (ca. 5 picograms) for the former and >50,000 fold more for the latter (ca. 200 nanograms). When collecting all cells from the cortex (e.g. Fig S2C), this material is sufficient for both. In contrast, when dealing with the FAC-sorted fraction of in utero electroporated cells (2-5% of the total; Fig. S4) this is no longer the case.

Author response to point of this reviewer regarding Figure 4 and 5:

We thank the reviewer for pointing out these aspects and have now i) added the levels of 5(h)mC/5hmC to Fig. 4C, ii) included a reference to Fig. 5 in the Discussion and iii) cited the two studies where this aspect first appears in page 7.

REVIEWER 3

We thank the reviewer for the interest in our work and many positive comments. We appreciate the constructive criticism and agree on the importance of validation. Concerning this, our original version included oxidative bisulfite sequencing of 6 loci and a total of 48 cytosines that in 44/48 (Fig. 2D) cases supported our data. We now provide two additional examples in Fig. S2. With regard to Fig. S2C, we agree that the changes in 5hmC between PP and DP are limited to a few nucleotides and that, considering the log scale, these changes are minor relative to the more substantial between DP and N. The critical analysis of this reviewer is to the point and well taken but as the reviewer mentions, this Fig. S2C represented just one example. In the new examples the reviewer may find other cases of “bivalency” of DP although it is true that the extent of “bivalency” depends on which particular nucleotide is being looked at. Clearly, our conclusions should not be based on a handful of ad-hoc examples but better deduced from unbiased bioinformatic analyses at the genome-wide level from which the “bivalency” of DP is discussed in our manuscript. Last, yes, nucleotides in Fig. S2C are few (3-4) and also the new examples shown in Fig. S2 are not many more (3-6). Yet, one nucleotide was claimed to be sufficient to alter gene expression (Furst et al., 2012) and even our manipulation of Dchs1 involving a few nucleotides (7-8) was sufficient to alter corticogenesis. We are unable to predict the significance of 5(h)mC based on nucleotide number and therefore have excluded this as a criterion from our analysis. Similarly, we are afraid that no method exists to resolve strand-specific asymmetries in DNA modifications.

Author response to point 1 of this reviewer:

We agree, it would be nice to have these data but given the limited amount of material that can be obtained from these cells in vivo we relied on previously published data (Bayam et al., 2015; Velasco et al., 2017, and the ENCODE project). Also, given the pressure to reduce the use of animal experiments we considered reproducing those published data not essential.

Author response to point 2 of this reviewer:

We thank the reviewer for the nice pick. Yes, upregulation of neuronal genes starts before the second differentiation step (DP-N) and can be detected, although as expected at lower level, in the first step (PP-DP) extending the concept of “bivalency” of DP. This is shown in Fig. 2F where a minor, yet highly significant ($p < 1e-7$ and $e-10$), upregulation of these genes can be seen. As described in our Discussion, we find this result intriguing because it is long known that certain neuron-specific genes (including markers such as bIII-tubulin, Map2 and others) start to be upregulated already in DP. Why and how these neuron-specific genes are expressed in progenitors, nobody knew. Our data may explain this for the first time.

Author response to point 3 of this reviewer regarding previous work done on TET knockout neural samples:

These studies were based on systemic knockouts leading to many diverse and long-term effects. Unfortunately, comparing this to our acute analysis in physiological conditions would not be informative as it would not allow us to dissect primary from secondary effects.

Author response to point 4 of this reviewer regarding figure 4c:

See also comment to Reviewer 2. Oxidative bisulfite conversion for 5hmC requires ca 50,000 fold more material than bisulfite sequencing for 5mC (ca 200 nano-g vs. 5 pico-g). A merit of our study is the in vivo analysis, an intrinsic caveat of which is the limited amount of cells that can be obtained. In Fig. S2 we could obtain enough material for both 5hmC and 5mC from FAC-sorted cells from the entire brain (this was in physiological conditions). In contrast, in Fig. 4C and S4 the amount of material was limited only to the small fraction of cells (ca. 2-5%) that have been targeted (this was after in utero electroporation). This is in fact one reason why previous studies overwhelmingly focused on cell lines in vitro.

February 18, 2019

RE: Life Science Alliance Manuscript #LSA-2019-00331-TR

Prof. Federico Calegari
DFG-Research Center and Cluster of Excellence for Regenerative Therapies, Faculty of Medicine,
Technische Universität Dresden
Fetscherstr. 105
dresden, Dresden 01307
Germany

Dear Dr. Calegari,

Thank you for submitting your revised manuscript entitled "Assessment and Manipulation of DNA (Hydroxy-)Methylation During Mammalian Corticogenesis". I appreciate your detailed point-by-point response and the introduced changes, and I would be thus happy to publish your paper in Life Science Alliance pending final revisions necessary to meet our formatting guidelines:

- when providing p-values, please also mention the statistical test used in the figure legends
- please note that files S1 and S2 were not uploaded
- please note that an 'A' is missing for figure legend S4A
- please note that the model in figure 5 is currently labeled as 'figure 6'.

A. FINAL FILES:

-- Summary blurb (enter in submission system): A short text summarizing in a single sentence the study (max. 200 characters including spaces). This text is used in conjunction with the titles of papers, hence should be informative and complementary to the title. It should describe the context

and significance of the findings for a general readership; it should be written in the present tense and refer to the work in the third person. Author names should not be mentioned.

B. MANUSCRIPT ORGANIZATION AND FORMATTING:

Sincerely,

February 20, 2019

RE: Life Science Alliance Manuscript #LSA-2019-00331-TRR

Prof. Federico Calegari
DFG-Research Center and Cluster of Excellence for Regenerative Therapies, Faculty of Medicine,
Technische Universität Dresden
Fetscherstr. 105
dresden, Dresden 01307
Germany

Dear Dr. Calegari,

Thank you for submitting your Research Article entitled "Assessment and Site-Specific Manipulation of DNA (Hydroxy-)Methylation During Mouse Corticogenesis". It is a pleasure to let you know that your manuscript is now accepted for publication in Life Science Alliance. Congratulations on this interesting work.

DISTRIBUTION OF MATERIALS:

Again, congratulations on a very nice paper. I hope you found the review process to be constructive and are pleased with how the manuscript was handled editorially. We look forward to future exciting submissions from your lab.

Sincerely,

Andrea Leibfried, PhD
Executive Editor
Life Science Alliance
Meyershofstr. 1
69117 Heidelberg, Germany
t +49 6221 8891 502
e a.leibfried@life-science-alliance.org
www.life-science-alliance.org